# Submarine slope failures due to pipe structure formation

Judith Elger[1], Christian Berndt [1], Lars Rüpke[1], Sebastian Krastel[2], Felix Gross[2] & Wolfram H. Geissler[3]

There is a strong spatial correlation between submarine slope failures and the occurrence of gas hydrates. This has been attributed to the dynamic nature of gas hydrate systems and the potential reduction of slope stability due to bottom water warming or sea level drop. However, 30 years of research into this process found no solid supporting evidence. Here we present new reflection seismic data from the Arctic Ocean and numerical modelling results supporting a different link between hydrates and slope stability. Hydrates reduce sediment permeability and cause build-up of overpressure at the base of the gas hydrate stability zone. Resulting hydro-fracturing forms pipe structures as pathways for overpressured fluids to migrate upward. Where these pipe structures reach shallow permeable beds, this overpressure transfers laterally and destabilises the slope. This process reconciles the spatial correlation of submarine landslides and gas hydrate, and it is independent of environmental change and water depth.

---

[1] GEOMAR Helmholtz Centre for Ocean Research, Wischhofstrasse 1-3, 24148 Kiel, Germany. [2] Institut für Geowissenschaften, Christian-Albrechts-Universität zu Kiel, Otto-Hahn-Platz 1, 24118 Kiel, Germany. [3] Alfred-Wegener-Institut Helmholtz-Zentrum für Polar- und Meeresforschung, Am Alten Hafen 26, 27568 Bremerhaven, Germany. Correspondence and requests for materials should be addressed to J.E. (email: jelger@geomar.de)

Landslides have the potential to generate tsunamis and destroy seafloor infrastructure. Spatial correlation between numerous submarine landslides and the occurrence of gas hydrates[1,2] suggests a causal relationship. The leading paradigm for more than a decade stated that the dissociation of gas hydrates destabilises continental slopes, because it removes the hydrate cementation and reduces the shear strength, while at the same time it increases overpressure due to gas expansion[3,4]. Although there is circumstantial evidence that hydrates may have an effect on the evolution of landslides[5], it was not possible to find unequivocal proof that any of the large submarine landslides were triggered by gas hydrate dissociation[6]. Hydrate is particularly sensitive to changes in pressure and temperature conditions close to the interception of the hydrate stability field and the seabed on the upper slopes[7,8]. The fact that many submarine landslides are retrogressive and originated at the middle or lower continental slope[9,10] contradicts the hypothesis that hydrate dissociation triggers slope failure. Other studies link submarine landslides to overpressure caused by an inversion of permeability[11,12] (Fig. 1), e.g., due to gas hydrates.

In this study, we investigate the feasibility of a new process that links gas hydrate systems and submarine mass wasting. The process combines various verified mechanisms that we put into a new context. Gas hydrates reduce the permeability of sediments[14,15] resulting in the accumulation of free gas and the buildup of overpressure below the gas hydrate stability zone (GHSZ)[16,17]. Elevated pore pressure may cause hydrofractures in the sediments, which in turn leads to pipe formation and transfer of overpressure to shallower coarse-grained sediments[18], to trigger slope failure. This novel mechanism of submarine slope failure initiation does not require any changes in the gas hydrate stability conditions and is applicable to all water depths.

The objective of this study is to constrain the environmental conditions under which this process is viable and to constrain the required parameters. We use seismic and hydroacoustic data from offshore N Svalbard showing a pipe structure reaching from the base of the GHSZ up to the base of a mass transport deposit, in combination with theoretical and numerical models of hydrofracture and overpressure generation, to evaluate (1) the required gas column height underneath the bottom-simulating reflection (BSR) to initiate hydrofracturing and pipe structure formation, (2) if it is feasible that a pipe may stop within the subsurface once it has started to propagate upwards, and (3) if overpressure may start to build up laterally within the subsurface to trigger a retrogressive submarine landslide.

## Results

**Seismic data.** Interpretation of two-dimensional (2D) seismic data from the rim of the Hinlopen Slide north of Svalbard (location in Supplementary Fig. 1) offers a different explanation for the link between the presence of gas hydrate and slope failure. A low seismic amplitude anomaly, i.e., blanking, in ~ 800 m water depth rises from 1.4 s two way travel time (twtt) corresponding to 215 m below sea floor (mbsf) to ~ 45 mbsf assuming an average seismic velocity of 1,700 m s$^{-1}$ (Fig. 2). Sedimentary reflections bend upwards around the anomaly that leads from an area of enhanced reflectivity to a body without visible internal structure (Fig. 2). We interpret it as a pipe structure of ~ 20 m diameter from a BSR up to the base of a mass transport deposit. The BSR marks the base of the GHSZ and mimics the seafloor. A broadening of the anomaly close to its upper termination indicates a reduction in confining pressure[19], suggesting that its top is at or close to the base of the mass transport deposit and that it was not significantly eroded by the landslide.

Gas hydrates reduce the permeability of sediments[14] resulting in the accumulation of free gas and the build-up of overpressure below the GHSZ[16,17]. High-amplitude seismic anomalies indicate a gas lens underneath the BSR of ~ 45 m height corresponding to a pore overpressure of at least 452 kPa (assuming interconnected gas pockets and a density contrast of ~ 1,024 kg m$^{-3}$ between water and gas). Considering a bulk density in the range of 1,690–2,140 kg m$^{-3}$ [20], this pressure corresponds to 58–73% of the overburden stress (Supplementary Table 1). The wavy reflections around the pipe structure (Fig. 2) indicate well-stratified sediment waves of clay and silt grain sizes[21] but could be biased by the 2D mapping.

**Overpressure calculation.** Elevated pore pressure may cause hydrofractures in the sediments, which in turn leads to pipe formation[22]. We propose that the associated transfer of overpressure to shallower coarse-grained sediments has triggered the slope failure. In order to test the feasibility of this scenario, we calculate the critical pore overpressure that initiates hydrofracturing or shear failure for a wide value range of static Poisson ratios, cohesions, and bulk densities at the bottom and top of the pipe structure (Table 1 and Fig. 2, parametric study in the methods; Supplementary Fig. 2). Cohesion is the most critical parameter but it is poorly constrained for marine sediments with estimates ranging between 0 and 2 MPa[23–26]. We consider a value of 280 kPa for the base of the GHSZ as the presence of gas hydrates[24] increase the stiffness and the shear strength of sediments. Our calculations indicate a critical pore overpressure of ~

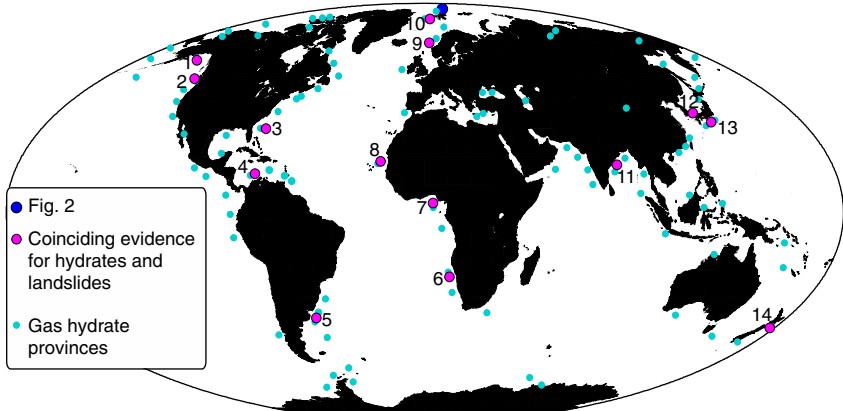

**Fig. 1** Global compilation of large submarine landslides in areas with gas hydrates. A global distribution of gas hydrate provinces (turquoise points[3,13]), coinciding evidence for hydrates and landslides (magenta dots) and the location of Fig. 2 (information on the references for the magenta data points in Supplementary Note 1)

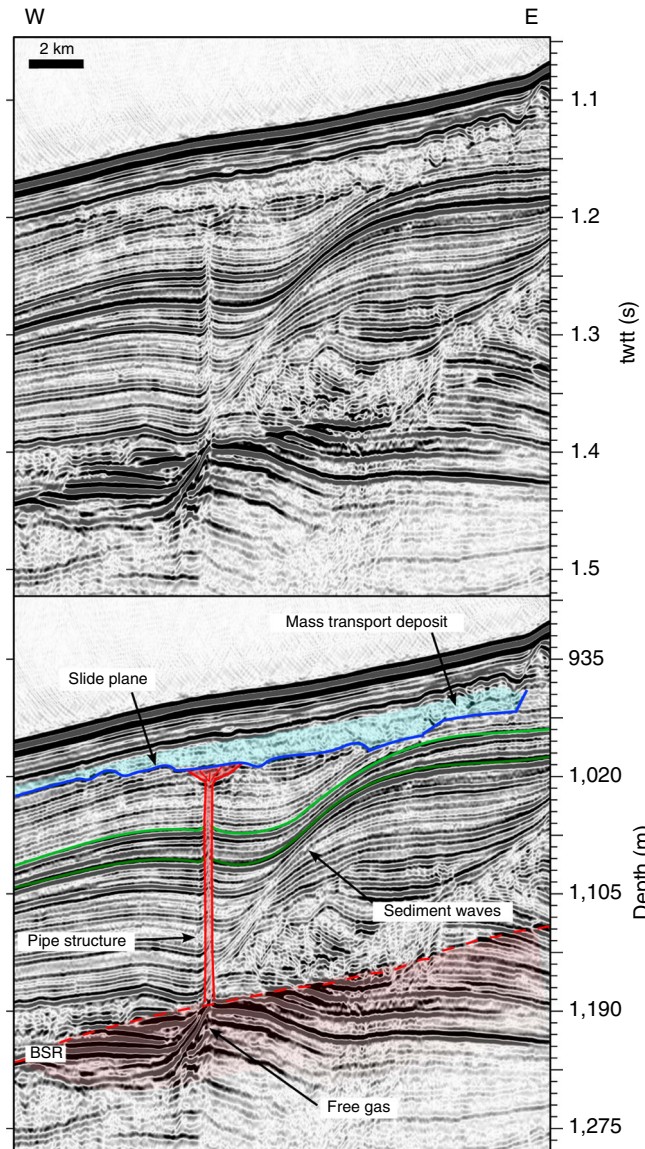

**Fig. 2** Reflection seismic profile and its interpretation with pipe structure and mass transport deposit. An extract of profile 20130390 without interpretation at the top (location in Supplementary Fig. 1) showing parallel reflections and bodies without visible internal structure, and its interpretation at the bottom showing stratified layers, headwalls (black lines), two highlighted colour-coded reflectors (green), the slide plane (blue) with the mass transport deposit (light blue), and a pipe structure reaching from the BSR to the mass transport deposit with the free gas below (red). The assumed sediment sound velocity for depth calculation is 1,700 m s$^{-1}$

890 kPa (786–1,619 kPa) to initiate tensile failure at ~ 215 mbsf (Fig. 3 and Table 1). At the top of the pipe structure (~ 45 mbsf), we assume negligible cohesion for normally consolidated marine sediments in the absence of gas hydrates[24] and up to 50 kPa for marine sediments with enhanced clay fraction (about 35% of the overburden stress)[25,26]. Critical pore overpressure for shear failure is in the order of 15–162 kPa for a range of static Poisson ratios, cohesion and friction angles (Table 1).

## Discussion

Previous studies of similar sedimentary environments showed that free gas can generate or reactivate fluid migration pathways

### Table 1 Parameters and results of critical overpressure calculation

| Depth (mbsf) | Failure mode | $\rho_{bulk}$ (kg m$^{-3}$) | $v$ | $\Phi$ (°) | $C$ (kPa) | $p_{crit}$ (kPa) | $h_{gas}$[a] (m) |
|---|---|---|---|---|---|---|---|
| 215 | Tensile | 1,800 | 0.3 | 30 | 280 | 916 | 91 |
| 215 | Tensile | 1,690[b] | 0.3 | 30 | 280 | 790 | 78 |
| 215 | Shear | 2,140[b] | 0.3 | 30 | 280 | 1,069 | 106 |
| 215 | Tensile | 2,140[b] | 0.37 | 30 | 280 | 1,653 | 164 |
| 45 | Shear | 1,690[b] | 0.3 | 25 | 50 | 162 | 16 |
| 45 | Shear | 1,690[b] | 0.3 | 25 | 0 | 15 | 1 |

[a] Gas column height supposing 100% replacement of water by gas
[b] Minimal per maximal bulk density from ODP 911A
Failure modes at 45 and 215 mbsf that result from different parameters (Poisson ratio $v$, friction angle $\phi$, Cohesion $C$, bulk density $\rho_{bulk}$) at a critical pressure $p_{crit}$ or gas column height $h_{gas}$ (assuming 100% replacement of water by gas)

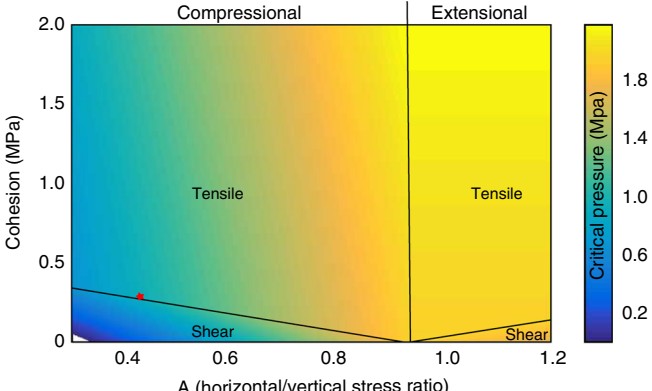

**Fig. 3** Critical pore overpressure as a function of horizontal per vertical stress ratio and cohesion at 215 mbsf. The graph shows the critical overpressure (colour coded) under extensional ($\sigma_V<\sigma_H$) or compressional ($\sigma_V>\sigma_H$) conditions that causes tensile or shear failure. The red dot represents critical pore overpressure for tensile failure at the bottom of the GHSZ

within the GHSZ in critically pressured systems[16,17,27]. Sedimentary structures may influence the build-up of overpressure by focusing fluid flow and forcing migration of fluids, e.g., in sediment waves or along faults (Fig. 2). The estimated pore pressure per overburden stress ratio of 58–73% (Supplementary Table 1) hints at a gas migration system at depth. Assuming similar conditions during pipe structure formation, it developed under critically pressured conditions. A gas column height of ~ 90 m in 215 mbsf initiates tensile failure and forms hydrofractures (Figs. 3 and 4a, and Table 1) but would be a conservative estimate assuming that overpressure originates from complete replacement of pore water by gas. Assuming gas lenses of bigger dimension suggest that this process is unrealistic. However, several studies observed such pipe structures in other study areas[19,22,28] and showed that mechanical compaction and fluid migration are capable to generate significant additional overpressures[29], especially at the presence of gas hydrates in clayey sediments[30]. Numerical simulations (Supplementary Fig. 3) of overpressure generation from mechanical compaction, loaded with the inferred sedimentation history of the Hinlopen area, predict pore overpressures of 30–380 kPa beneath the GHSZ for hydrate saturations of 20–60%, assuming no lateral fluid migration. This reduces the required critical gas column height by 3–38 m. We conclude that the combination of buoyancy and other sources of

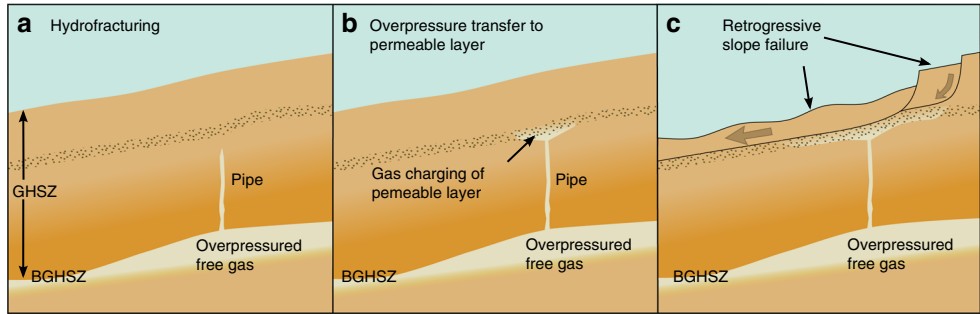

**Fig. 4** Schematic evolution of retrogressive slope failure due to overpressured gas below the GHSZ: **a** submarine slope with gas hydrate-bearing sediments (gradually decreasing saturation from the bottom of the GHSZ) and overpressured gas (bright area) at the bottom of the GHSZ (BGHSZ) induces pipe generation into the GHSZ, **b** the conduit encounters a permeable layer; gas enters and leads to overpressure transfer from the bottom of the GHSZ to the shallow subsurface, **c** overpessured gas causes shear banding in the weak layer and generates retrogressive slope failure

pore pressure, e.g., mechanical compaction during sedimentation, generated critical gas overpressure.

Once hydrofracturing occurs, a pipe structure may develop and transfer overpressure to layers in the shallow subsurface (Fig. 4a, b). Within a fast-forming conduit pore water, free gas and dissolved gas would start to migrate upwards, maintaining the pressure of the overpressure reservoir as long as the reservoir is large compared with the fracture volume in the pipe structure. Free gas can migrate through the GHSZ without forming gas hydrate due to limited water supply[31], shifting pressure–temperature conditions[32], capillary effects in fine-grained sediments[31], high pore water salinity[33] or a combination of these[34]. Ample evidence for gas seepage through the GHSZ has been observed throughout the world[35,36]. The precise mechanism for gas migration through the GHSZ is of secondary importance as long as hydrate formation does not close the conduit.

In order to transfer the pore pressure from the overpressure zone at the base of the GHSZ into the shallow subsurface, a developing pipe structure must not propagate to the sea floor but bleed overpressure into a shallow sediment layer of higher permeability than the overlying beds. This process is favoured by heterogeneity and anisotropy of the penetrated material, e.g., discontinuities, stress barriers or layers of strongly contrasting Young's moduli[37]. Dense layers can function as stress barriers and encourage arrest of fractures[37], whereas layers with a high fraction of sand increase the permeability and facilitate horizontal fluid migration. Indications for such barriers are seismic stratification (Fig. 2) and local density differences of ~ 0.4 g cm$^{-3}$ in Ocean Drilling Program (ODP) bore hole 911 A[20]. In contrast to clay-rich sediment, cohesion of normally consolidated shallow sediments is negligible[24,25] and forces failure to change from tensile to shear (Fig. 3), which is shown in seismic data and sand box models[19]. The calculated critical pore overpressure at ~ 45 mbsf is on the order of 15–160 kPa (Table 1). This is consistent with the observed broadening of the pipe in its upper part as resistance to sideway propagation diminishes (Fig. 4b). Once deformation takes place and the weak zone exceeds a critical length, a shear band may propagate into the adjacent slope and can initiate retrogressive slope failure (Fig. 4c)[38,39]. As we assume fluid migration in a rapidly forming conduit we do not consider gas hydrate formation, as the system will be out of thermal equilibrium in the shallow sediments. The transfer of pressure through inter-pore connections and by water migration, i.e., without free gas, would be other possible scenarios.

A requirement for the process to work is the presence of a permeability inversion in the shallow subsurface, otherwise a developing pipe structure will propagate all the way to the seabed such as the pipe structures that have formed next to the Storegga

Slide[22]. However, weak layers consisting of sands and permeability inversions are frequently encountered in geotechnical studies that investigate submarine slope stability[39].

Although it is conceivable that other causes for permeability inversions may start pipe formation also at shallow sub-bottom depth, ubiquity of pipe structures in hydrate provinces[36] show that permeability changes due to gas hydrate formation is a particularly efficient process. The straight and vertical limit of pipe structures indicates that they form quickly and onlap of the adjacent reflectors onto the upwarped reflectors inside the pipe structure at distinct stratigraphic intervals suggests periodicity in their formation[22]. Therefore, pipe structure formation provides a re-occurring destabilisation mechanism and trigger process that explains why there is no clear clustering of landslide ages for specific times of the climate cycle. These implications have to be considered in geohazard risk assessments for continental margins with natural gas hydrate systems.

## Methods

**Geophysical data**. The study bases on a time-migrated ~ 22.6 km-long 2D high-resolution seismic profile (20130390) using a digital 80-channel Geometrics GeoEel streamer of 125 m total length and a group spacing of 1.5625 m from the NE slope of the Hinlopen-Yermak Slide (Fig. 2 and Supplementary Fig. 1). A 1.7 l SERCEL GI air gun was shot in harmonic mode at 200 bar in ~ 2 m water depth[40]. Seismic processing was carried out by using the commercial software Schlumberger Vista Seismic Processing 13. Data were sampled at 0.5 ms and sorted into common midpoint domain with a bin spacing of twice the group spacing. Normal move-out correction was applied with a velocity of 1,500 m s$^{-1}$ and an Ormsby bandpass filter with corner frequencies at 10, 20, 200, and 400 Hz. Due to the short length of the streamer system and relatively great water depths, the data were time migrated with water velocity, as the streamer offset is not long enough for velocity analysis. The shooting intervals of 7 s at ~ 4.5 knots results in a shot point distance of 16 m. The entire water column was recorded during seismic acquisition. The seismic profile starts at 81°04.163′N/17°17.735′E and ends at 81°04.209′N/15°54.633′E (see Supplementary Fig. 1).

Multibeam bathymetric data were recorded during MSM31 using the hull-mounted Kongsberg Simrad EM122 system with 191 beams per ping, an angular coverage of 150° and 12 kHz nominal frequency[35]. Bathymetric data were processed using the software CARIS HIPS & SIPS and gridded with GMT. The grid shown in Supplementary Fig. 1 has a horizontal bin size of 50 m.

**Overpressure calculation**. The calculation of critical pore pressure is based on the Mohr–Coulomb criterion for shear failure[41] and the theoretical criterion for tensile failure of a fluid-filled crack from Griffith's theory[42]. The methodology of inferring the failure mode induced by a localised fluid overpressure source under different initial stress states is taken from ref. [43], using only failure mode one to four. These equations lead to a definition of the critical pore-fluid overpressure for shear and tensile failure in different compressional and extensional regimes.

To determine the static Poisson ratio $\nu$ we assume that the maximum total stress ($\sigma_1$) is vertical, and that only compaction and no significant tectonic forces are present. We assume a coefficient of earth pressure at rest, defined as the horizontal to vertical stress ratio[36], of 0.42–0.7 for coarse sand to compacted sand in layers, respectively[37]. The relationship $\sigma_2 = \sigma_3 = \frac{\nu}{(1-\nu)}\sigma_1$, with horizontal stress tensors $\sigma_2$ and $\sigma_3$, lead to static Poisson ratios $\nu$ of 0.3–0.4. To calculate the ratio of pore and lithostatic pressure, we used bulk densities from ODP hole 911 A[38] to

calculate the lithostatic pressure and define the pore pressure as the sum of hydrostatic pressure and overpressure.

**Ratio of pore pressure to lithostatic pressure**. We calculated the ratio of pore pressure to lithostatic pressure in Supplementary Table 1 by the equations (1) and (2) with gravitational acceleration $g$, depth $-y$, gas column height $h_{gas}$ and density $\rho$ (assuming interconnected gas pockets and a density contrast of ~ 1,024 kg m$^{-3}$ between water and gas).

$$p_{\text{lithostatic}} = \rho_{\text{bulk}} \times g \times y \qquad (1)$$

$$p_{\text{pore}} = p_{\text{hydrostatic}} + p_{\text{overpressure}} = (\rho_{\text{water}} - \rho_{\text{gas}}) \times g \times y + (\rho_{\text{water}} - \rho_{\text{gas}}) \times g \times h_{\text{gas}} \qquad (2)$$

**Parametric study**. The equations that determine the critical overpressure and the failure mode depend on a number of geotechnical parameters of the subsurface and on assumptions on the stress regime. Due to the lack of drill sites most of the parameters were taken from literature or estimated from similar studies and thus have associated uncertainties. In order to assess the influence of each parameter on the critical overpressure and the failure mode, we calculated the critical overpressure as a function of each parameter and the coefficient of earth pressure at rest for 45 and 215 mbsf. The values for the constant parameters are listed in Supplementary Table 2. The parametric study focusses on a coefficient of earth pressure at rest of 0.4 to 0.7 in agreement with several studies[44,45].

Cohesion of marine sediments is poorly constrained. We adopt the estimates in the literature that range between 0 and 2 MPa[23,24,26,46]. Supplementary Fig. 2 shows the critical pressure as a function of cohesion and stress ratios in 45 and 215 mbsf. Increasing cohesion causes a shift from shear to tensile failure and increases the critical pressure for failure. At 45 mbsf, the zone characterised by cohesion and stress ratio that causes shear failure is very small. As these sediments in the shallow subsurface are less consolidated and the amount of gas hydrates should be negligible, cohesion is most likely negligible (c.f., refs. [24,47]) and shear failure is the most likely failure mode. These constrains limit the critical pressure for failure to maximal ~ 0.2 kPa, a restricted area in Supplementary Fig. 2. Cohesion of sediments at the bottom of the GHSZ in 215 mbsf is not negligible, because they underwent mechanical compaction and contain relevant amount of gas hydrate. Supplementary Fig. 2 shows that cohesion is critical to determine the failure mode and allows a wide range of critical pressure for a particular stress ratio that corresponds to about 40 m of gas column height (supposing 100% replacement of water by gas).

We set the static Poisson ratio to a value of 0.3 following other studies, e.g., ref. [48]. In order to take into account that this is only a rough estimate, and that there are other studies that propose the static Poisson ratio to range between 0.41 and 0.45[17], we have calculated the critical pressure for failure as a function of stress ratio and static Poisson ratio ranging between 0.25 and 0.45 in 45 and 215 mbsf (Supplementary Fig. 2). Regardless of the depth, the static Poisson ratio does not change the failure mode under most pressure conditions. Increasing static Poisson ratio decreases slightly the critical pressure.

Values for bulk density of the subsurface in the calculations range from 1,690 to 2,140 kg m$^{-3}$ according to results from IODP site 911 A[20]. For sediments at 45 mbsf, the effect of density on critical pressure and failure mode is negligibly small and there is no influence on the failure mode (Supplementary Fig. 2). At the bottom of the GHSZ (215 mbsf), changes of density can lead to a change from tensile to shear failure with increasing density (Supplementary Fig. 2). Increasing density of the material requires an increasing critical pressure for tensile failure.

Several studies estimated the friction angle of marine sediments to 30° [48]. Using the correlation of the coefficient of earth pressure at rest $K_0$ (horizontal to vertical stress ratio, ranging from 0.42 to 0.7) and the friction angle $\phi$ ($K_0 = 1 - \sin\phi$) from ref. [49], we tested friction angles between 10° and 40°. Supplementary Fig. 2 shows that very small friction angles prevent failure in the shallow subsurface. Increase of friction angle correlates with negligible or no increase of critical pressure in 45 and 215 mbsf.

The Biot-Willi poroelastic coupling constant varies in our calculations between 0.67 and 0.77, and corresponds to 25–0% bulk hydrate in sediment[17]. Supplementary Fig. 2 shows that there is no influence on the failure mode and negligible influence on the critical pore pressure at the bottom of the GHSZ and in the shallow subsurface.

The horizontal to vertical stress ratio, or coefficient of earth pressure at rest[49], has a significant influence on the failure mode and on the critical pore pressure in every depth (see Supplementary Fig. 2). In agreement to values in the literature, we chose the range from 0.42 to 0.7, which describes coarse sand to compacted sand in layers, respectively[45].

**Overpressure from sediment compaction**. In order to assess the critical gas column height necessary to initiate hydrofracturing, the background pressure state has to be known. Sediment compaction can raise the pore fluid pressure well above hydrostatic values. Here we investigate under-compaction and overpressure generation beneath the GHSZ. For this purpose, we solve the pressure equation (3) to

(5):

$$\frac{\mathrm{D}\phi}{\mathrm{D}t} = -C \frac{\mathrm{D}\sigma'_z}{\mathrm{D}t} \qquad (3)$$

$$\sigma'_z = u_1 - u = p_{\text{lithothatic}} - p_{\text{hydrostatic}} - u \qquad (4)$$

$$\frac{C}{(1-\phi)}\frac{\mathrm{D}u}{\mathrm{D}t} - \nabla \cdot \left(\frac{k}{\mu}\nabla\mathrm{u}\right) = \frac{C}{(1-\phi)}\frac{\mathrm{D}u_1}{\mathrm{D}t} \qquad (5)$$

where $u$ is the excess pore pressure ($p - p_{\text{hydrostatic}}$), $C=0.09613 \times 10^{-6}$ Pa$^{-1}$ is the Athy compaction constant, $k$ is permeability, $\phi$ is the friction angle and $\mu$ is the temperature-dependent viscosity. We use a porosity-dependent permeability function for shales[50] and assume that the permeability further changes with hydrate saturation $S_h$ within the GHSZ[34]:

$$k = k_0\left[1 - S_h^2 + \frac{2(1-S_h)^2}{\log(S_h)}\right] k = k_0\left[1 - S_h^2 + \frac{2(1-S_h)^2}{\log(S_h)}\right] \qquad (6)$$

The above equations have been implemented in Matlab using a Lagrangian finite element method. The model is initialised to a 1,140 m-thick layer that is at hydrostatic conditions. Sediments (360 m) are added in each simulation throughout the Quaternary at deposition rates inferred for the study area[20] so that the final sediment thickness is 1,500 m. Supplementary Fig. 3 shows the results of three example calculations assuming 20%, 50% and 60% hydrate saturation (c.f. wide range of hydrate saturation, e.g., in refs. [51,52]) within the GHSZ (cf., ref. [52]). For these three model runs, overpressures of 30, 170 and 380 kPa are predicted at the base of the GHSZ, which reduces the critical gas column heights by 3–38 m.

**Data availability**. The data sets analysed during the current study are available from the corresponding author on reasonable request.

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

## Acknowledgements

We thank the captain of R.V. Maria S. Merian and his crew for their excellent support at sea. Ship time for cruise MSM31 was provided by the DFG Senatskomission für Ozeanographie. J.E. was financed by the Helmholtz graduate school HOSST.

## Author contributions

J.E., C.B., and S.K. conceived the idea for this study. L.R. and J.E. performed the numerical modelling. F.G. and W.H.G. acquired and processed the seismic data. J.E. wrote the first draft of the paper and all co-authors improved the final version of the manuscript through multiple iterations of the text and figures.

## Additional information

**Competing interests:** The authors declare no competing financial interests.

