## [Peer Review File · Nature Communications]

Reviewers' comments:

Reviewer #1 (Remarks to the Author):

The manuscript "Submarine slope failures due to pipe structure formation" by Elger et al. proposed a novel mechanism for submarine slope failures. The topic of this paper is suitable for the journal, and the whole paper is well written. However, a few problems are spotted which may help improve the quality of this MS. It is also probably just because the reviewer didn't understand the point, and please explain that in detail to make it easier to understand. Firstly, the proposed mechanism is quite novel, but other possibilities should not be excluded, right? As I see, different areas may have different mechanisms for slope failures. For example, in areas without pipe formation at the shallower area near the landward limit of the GHSZ, can the overpressure caused by dissociated gas lead to slope failures? Secondly, the transfer of overpressure is realized by fluid migration along the pipe in this case. It is believed that there is not enough water for hydrate formation within the pipe. But will the free gas form hydrates at the shallower permeable reservoir which is also located in the GHSZ? Additionally, it seems that the pipe bottom is consistent with the onlapping line of the sediment wave. Does the sediment wave play a role in the location of pipe formation? e.g. due to the weak geophysical properties. To summarize, I think it is an interesting paper and worthy to be published although a few points need to be clarified. Some other major comments are listed below, and please see attached PDF file for all the comments, changes and suggestions I have made.

1. The pipe does not reach the seabed in this study. Is it worth discussing the factors that control the pipe length: overpressure of the free gas column below the BSR, physical properties of sediments, etc.?

2. Can you explain the stress condition at the pipe in detail, and show the specific equations in the Methods part, other than in the supplementary information?

3. Is pipe formation process episodic? But reactivated several times? How about the sliding process, gradually or episodic?

Reviewer #2 (Remarks to the Author):

This paper aims to show that overpressure buildup at the base of the gas hydrate stability zone can result in hydro-fracturing of the overlying sediment, which then forms a pipe structure that propagates upward, providing pathways for overpressured fluids to migrate upward, spread laterally through a shallow permeable layer and destabilize the overlying strata, thus destabilizing the slope and leading to landsliding. The links between pipe structures and BSR and between the pipes and the base of an MTD look very reasonable in this specific area.

This is indeed a novel concept that will be of interest to the hydrate-landslide community as well as the wider field of continental margin sedimentation/tectonics. The conclusions are original and are well-supported by data and model studies. This paper should stimulate thinking about the causes of submarine landslides in gas hydrate provinces around the world.

The paper is very well-written and organized. I have only a few (very minor) questions and suggestions to improve the English:

Line 50 – should be "...water depths..."

Line 80 – If pipe extends toward surface, wouldn't hydrostatic pressure decrease, which would in turn cause overpressure to diminish?

Line 98 – isn't there also a zone of enhanced reflections, colored in red?

Line 105 – maybe should use "at", rather than "in": "...at 45 and 215 mbsf..."

Line 120 – maybe should be “pathways”?

Line 154, similar to comment above, maybe should use “at”: “...at ~45 mbsf...”

Line 154, maybe use “on” instead of “in”: “...on the order...”

Lines 156-157, I don’t understand where it comes from: “...a shear band may propagate...” Please add more explanation about why “shear band”.

Line 167, “...overlap on up bend reflectors...” – this is a bit awkward, can you clarify this?

Gregory Moore
University of Hawaii

Changes

Reviewer #1

The manuscript “Submarine slope failures due to pipe structure formation” by Elger et al. proposed a novel mechanism for submarine slope failures. The topic of this paper is suitable for the journal, and the whole paper is well written. However, a few problems are spotted which may help improve the quality of this MS. It is also probably just because the reviewer didn’t understand the point, and please explain that in detail to make it easier to understand. Firstly, the proposed mechanism is quite novel, but other possibilities should not be excluded, right? As I see, different areas may have different mechanisms for slope failures. For example, in areas without pipe formation at the shallower area near the landward limit of the GHSZ, can the overpressure caused by dissociated gas lead to slope failures? Secondly, the transfer of overpressure is realized by fluid migration along the pipe in this case. It is believed that there is not enough water for hydrate formation within the pipe. But will the free gas form hydrates at the shallower permeable reservoir which is also located in the GHSZ? Additionally, it seems that the pipe bottom is consistent with the onlapping line of the sediment wave. Does the sediment wave play a role in the location of pipe formation? e.g. due to the weak geophysical properties. To summarize, I think it is an interesting paper and worthy to be published although a few points need to be clarified. Some other major comments are listed below, and please see attached PDF file for all the comments, changes and suggestions I have made.

1. The pipe does not reach the seabed in this study. Is it worth discussing the factors that control the pipe length: overpressure of the free gas column below the BSR, physical properties of sediments, etc.?

In line 147-154 we discuss potential factors that control the pipe length as “heterogeneity and anisotropy of the penetrated material, e.g. discontinuities, stress barriers or layers of strongly contrasting Young’s moduli”. A change from clay-rich to normally consolidated sediment would be one possible scenario that would force failure to change from tensile to shear. For a more detailed answer see our detailed response on the reviewer’s question to line 56-57 below.

2. Can you explain the stress condition at the pipe in detail, and show the specific equations in the Methods part, other than in the supplementary information?

We have moved the discussion of this question from the methods in the supplementary information to the method part in main manuscript that describes the overpressure calculation. The stress condition is not well constrained. We assume that we deal with a “normal” sedimentary basin with the maximum total stress in vertical direction, with compaction and no significant tectonic forces to be present (see method section).

3. Is pipe formation process episodic? But reactivated several times? How about the sliding process, gradually or episodic?

As we discuss in the paper (lines 167f), there are indications that the pipe formation process was rapid and episodic. The sharp vertical boundary of the conduit indicates rapid formation and the onlap of adjacent reflectors onto the up-warped chimney reflectors at discrete stratigraphic levels indicates episodic formation. In the discussed scenario, destabilization occurs instantaneously and only once the conduit reaches a shallow, permeable layer and transfers a critical pressure for failure.

It is conceivable that a long-lived pipe structure may trigger several landslides but we do not discuss this in the paper to avoid confusion.

Line 3-11: author from the same affiliation use the same number?

We have deleted the email addresses except the one of the corresponding author (line 11) and summarized the postal addresses.

Line 15: insert solid prior to supporting

Done

Line 35: replace “cementation by hydrates” with “hydrate cementation”

Done

Line 38-40: A Paper by Graves et al. (2017) talks about this position-landward limit of the GHSZ: "Methane in shallow subsurface sediments at the landward limit of the gas hydrate stability zone offshore western Svalbard"

We thank the reviewer for suggesting this literature. However, as that paper refers to the source of gas seepage on the margin West of Svalbard it is not too relevant as our study area north of Svalbard is several hundred kilometers away and the source of methane is a different one. Therefore, we did not include it in the text.

Line 47: GHSZ first appears here. It should be like this: below the gas hydrate stability zone (GHSZ)

Done

Line 49-50: quite novel. But other possibilities should not be excluded, right? As I see, different areas may have different mechanisms for slope failures.

Correct. Our intention is not to exclude any other processes with this paper. The goal is to introduce a new process that fits the spatial correlation of gas hydrate provinces and slope failures. In contrast to the hypothesis of gas hydrate dissociation, this processes is not limited to water depth where the gas hydrate stability zone crops out on the sea floor.

Line 56-57: does this mean the pipe does not reach the seabed? Is it worth discussing the factors that control the pipe length: overpressure of the free gas column below the BSR, physical properties of sediments, etc.

Correct, it means that the pipe does not reach the sea floor. The pipe needs to stop propagating in the subsurface to transfer the pressure from below the gas hydrate stability zone to the shallow subsurface. This can happen via inter-pore connection or via fluid (water or gas) migration laterally into coarser sediment and then initiates slope failure. In line 147-154 (line numbers of the first version of the manuscript) we discuss potential factors that control the pipe length as “heterogeneity and anisotropy of the penetrated material, e.g. discontinuities, stress barriers or layers of strongly contrasting Young's moduli”. A change from clay-rich to normally consolidated sediment would be one possible scenario that would force failure to change from tensile to shear.

Line 61: insert “Interpretation of” prior to “2D seismic data”

Done and adjusted the grammar (“offers”).

Line 81-82: The transfer of overpressure is realized by fluid migration, right? Will the free gas forms hydrates at the shallower permeable reservoir which is also located in the GHSZ? Is there enough water for hydrate formation?

Yes, the fluid transfers the overpressure to the shallow subsurface. We assume a fast forming conduit and therefore a relatively quick transfer of the fluid to the shallow subsurface and do not expect that gas hydrate would form under these conditions. (Even if there is enough water in the coarser shallow sediment gas would have to mix and dissolve quickly in the water which would require a rather slow process.)

Apart from that, we discuss a transfer of pressure. As soon as there is a connection of the overpressured fluid below the gas hydrate stability zone through the conduit to the fluids in the shallow pores of the shallow sediment (inter-pore connection), the pressure could be transferred without actual fluid migration. With fluids, we refer not only to gas but also to water. To clarify this, we address it later in the paper in the discussion part (see answer to reviewer's comment in line 138-139).

Line 94 (figure 2): Can you explain the stress condition in this location, and show the specific equations in the Methods part? Besides, do you consider the effect of the sediment wave on the pipe location? It seems that the pipe bottom is consistent with the overlapping line of the sediment wave where may have weak sediments?

The stress condition is not well constrained. We assume that we deal with a "normal" sedimentary basin with the maximum total stress in vertical direction, with compaction and no significant tectonic forces to be present (see method section). As suggested by the reviewer and the editor, we have moved the equations and some other information on the methods from the supplementary information to the main manuscript.

Yes, we have thought about the position of the sediment wave and the spatial correlation with the pipe structure. We inserted a sentence in line 120 to address the possible effect of the sedimentary structures on slope stability. But we do not feel confident to make final statement about the geometry below the pipe structure, because this is a three-dimensional structure i.e. the potential sediment wave, which would influence fluid migration in three dimensions. With the 2D seismic data it is impossible to determine if it crosses at the apex of the sediment wave. We have inserted a small comment in line 82.

Line 105: delete "that"?

We have kept "that" but adjusted "mode" to "modes".

Line 121: Is this due to the free gas zone below the BSR? How can it tell it is an active gas migration system?

Yes, the pore pressure ratio relates to the zone below the BSR, below the gas hydrate stability zone, where free gas exists. But we refer to fluids in general, not only gas. The gas can lead to overpressure due to buoyancy.

We have deleted "active" in the text, because it is true that we cannot confirm if the gas migration system is active at the moment. Geissler et al. (2016) show that there are several active seeps in the region, but they are in about 30 km distance.

Line 138-139: Within the pipe there is no enough water for hydrate formation, right? Will the free gas forms hydrates at the shallower permeable reservoir there is probably enough water for hydrate formation. How will the pore pressure change during the fluid charging process?

There are different possible reasons why gas hydrate does not form in the fast forming conduit. Limited water supply, shifting pressure-temperature conditions within the conduit, capillary effects in fine-grained sediments, high pore water salinity, or a combination of these processes are possible (line 138-140). As we explain in the comment to line 81-82, we do not expect gas hydrates to form in the shallow sediment and we assume fast gas migration. We have inserted an explanatory remark (line 157 ff.).

Line 172 (figure 4): in Fig. 4B, no hydrate forms in this stage? Pipe formation process is episodic, right? But the sliding process may be not?

No, we do not expect hydrate to form at this stage (see comments on questions to line 138-139).

The structure of the pipe and the geometry of the seismic reflections suggest that the pipe formation might have been episodic, but that is not necessary for the process (see also our response to comment 3 above).

All geological and geophysical observations suggest that the landslide moved in a single, short-lived event. As there is still gas in the subsurface which may result in a new episode of pipe propagation in the future, the same pipe may trigger another landslide in the future.

Reviewer #2

line 50: should be “...water depths...”

Done

Line 80: If pipe extends toward surface, wouldn't hydrostatic pressure decrease, which would in turn cause overpressure to diminish?

The overpressure is the difference between pore pressure and the hydrostatic pressure, and the reviewer is correct that the hydrostatic pressure decreases in the pipe towards the seafloor. However, the pore pressure does not decrease due to gas leakage into the surrounding rocks or depletion of the gas reservoir. This is the central assumption of this paper and corroborated by the seismic observations (fast pipe propagation, large gas reservoir, etc.). Therefore, the overpressure will increase in the pipe structure from the gas reservoir towards the shallow subsurface where it may trigger slope failure.

Line 98: isn't there also a zone of enhanced reflections, colored in red?

Correct. We have inserted “”, and a pipe structure reaching from the BSR to the mass transport deposit with the free gas below (red).”

Line 105: maybe should use “at”, rather than “in”: “...at 45 and 215 mbsf...”

Done

Line 120: maybe should be “pathways”?

Done

Line 154: similar to comment above, maybe should use “at”: “...at ~45 mbsf...”

Done

Line 154: maybe use “on” instead of “in”: “...on the order...”

Done

Lines 156-157: I don't understand where it comes from: “...a shear band may propagate...” Please add more explanation about why “shear band”.

We forgot to insert a reference. This is fixed now. We are now referring to the paper of Puzrin et al. (2015), that states that if a weak zone exceeds a critical length it transforms into a shear band that can cause the adjacent slope to fail as well. In order to better explain where “shear band” is coming from, we have inserted “and the weak zone exceeds a critical length” and updated the reference.

Line 167, “...onlap on up bend reflectors...” – this is a bit awkward, can you clarify this?

We have rephrased it to “onlap of the adjacent reflectors onto the up-warped reflectors inside the pipe structure at distinct stratigraphic intervals”.

Other changes:

Line 13: We have deleted the reference to Fig.1 and the supplementary information) due to the format guidelines of Nature Communications.

Line 15: We have deleted the references 1 and 2 to match the guidelines of Nature Communications.

Line 15-16: We have deleted the sentence “Quite to the contrary, initiation of some landslides in deep water makes this an unlikely scenario.” to shorten the abstract and match the word limit.

Line 15: We have deleted the reference 3 to match the guidelines of Nature Communications.

Line 22-25: We have rephrased this part of the abstract to “This process reconciles the spatial correlation of submarine landslides and gas hydrate, and it is independent of environmental change and water depth.” to shorten the abstract and match the word limit.

Line 27 (figure 1): We have added two other examples for coinciding evidence for hydrates and landslides (now point 2 and 12), and changed the colors from red to magenta and from green to turquoise. These were adapted in the line 30 and in the supplementary information in the references for the data points.

Line 28: Changed “gas hydrate” to “gas hydrates”.

Line 32: inserted “Introduction”

Line 39: deleted “special”

Line 61-93: We have created subheadings for the results of the seismic data and the overpressure calculation.

Line 94 (figure 2): We have included the not interpreted data in the top part of the figure and adjusted the caption below to refer to both parts of the figure. We deleted Extended Data Figure 4. We moved the figure to the subsection “geophysical data”.

Line 159: We have deleted the heading “Conclusion” as this is not part of the structure in Nature Communications.

Line 172 (figure 4): Changed BSR to BGHSZ (bottom of gas hydrate stability zone) and adjusted the caption of the figure.

Line 181-188: Inserted subheadings in the method section for geophysical data and the overpressure calculation and merged the information from the supplementary information and the main manuscript.

Line 275: We have changed the link so that it refers to the web page of nature communications.

References:

As we have deleted the references in the abstract the numbers that refer to the cited papers have changed.

We have added references to the work of Kretschmer et al. (2015) in line 40, to Dugan & Flemings (2000) in line 120, Karstens & Berndt (2015) in line 126, Liu & Flemings (2006) in line 140, and Puzrin et al. (2015) in line 157.

Figure Legends:

We have listed all figure captures at the end of the manuscript.

Supplementary Information

Line 38: We have moved Extended Data Figure 1 to this line and renamed it to Supplementary Figure 1. We adjusted all references in the main text and in the supplementary information.

Line 139: We have moved Extended Data Figure 2 to this line and renamed it to Supplementary Figure 2. We adjusted all references in the main text and in the supplementary information.

Line 164: We have moved Extended Data Figure 3 to this line and renamed it to Supplementary Figure 3. We adjusted all references in the main text and in the supplementary information.

References: We have updated the references.